



# Observations of the Lower Atmosphere From the 2021 WiscoDISCO Campaign

Patricia A. Cleary[1], Gijs de Boer[2,3,4], Joseph P. Hupy[5], Steven Borenstein[4], Jonathan Hamilton[2,3], Ben Kies[1], Dale Lawrence[6], R. Bradley Pierce[7], Joe Tirado[1], Aidan Voon[1], Timothy Wagner[7]

[1] Department of Chemistry, University of Wisconsin Eau-Claire, Eau-Claire, WI, 54701, USA
[2] Cooperative Institute for Research in Environmental Sciences, University of Colorado Boulder, Boulder, CO, 80309, USA
[3] Physical Sciences Laboratory, National Oceanic and Atmospheric Administration, Boulder, CO, 80305, USA
[4] Integrated Remote and In Situ Sensing, University of Colorado Boulder, Boulder, CO, 80309, USA
[5] School of Aviation and Transportation Technology, Purdue University, West Lafayette, IN, 47907, USA
[6] Research and Engineering Center for Unmanned Vehicles, University of Colorado Boulder, Boulder, CO, 80309, USA
[7] Space Science and Engineering Center, University of Wisconsin Madison, Madison, WI, 53706, USA

*Correspondence to*: Patricia A. Cleary (clearypa@uwec.edu)

**Abstract.** The meso-scale meteorology of lake breezes along Lake Michigan impacts local observations of high ozone events. Previous manned aircraft and UAS observations have demonstrated non-uniform ozone concentrations within and above the marine layer over water and within shoreline environments. During the 2021 Wisconsin's Dynamic Influence of Shoreline Circulations on Ozone (WiscoDISCO-21) campaign, two UAS platforms, a fixed-wing (University of Colorado RAAVEN) and a multirotor (Purdue University DJI M210), were used simultaneously to capture lake breeze during forecasted high ozone events at Chiwaukee Prairie State Natural Area in southeastern Wisconsin from May 21-26, 2021. The RAAVEN platform (data DOI: 10.5281/zenodo.5142491) measured temperature, humidity, and 3-D winds during 2-hour flights following two separate flight patterns up to 3 times per day at altitudes reaching 500 m above ground level. The M210 platform (data DOI: 10.5281/zenodo.5160346) measured vertical profiles of temperature, humidity and ozone during 15-minute flights up to 6 times per day at altitudes reaching 120 m above ground level (AGL) near to a WI-DNR ground monitoring station (AIRS ID: 55-059-0019). This campaign was conducted in conjunction with the Enhanced Ozone Monitoring plan from WI-DNR that included Doppler lidar wind profiler observations at the site (data DOI:10.5281/zenodo.5213039).



## 1. Introduction

The WiscoDISCO-21 (Wisconsin's Dynamic Influence on Shoreline Circulations on Ozone) was designed to capture lake breeze influence on shoreline ozone observations and to interrogate the low-altitude dimensionality of the marine layer as it moves on shore. The influence of lake breeze on shoreline air quality along Lake Michigan (Keen and Lyons, 1978; Lyons and Cole, 1976; Lyons and Olsson, 1973; Dye et al., 1995; Foley et al., 2011; Stanier et al., 2021) and other Great Lakes (Hayden et al., 2011; Levy et al., 2010; Wentworth et al., 2015; Sills et al., 2011) is well documented by campaigns incorporating ground (Lyons and Cole, 1976), ferry (Lennartson and Schwartz, 2002; Cleary et al., 2015) and aircraft observations (Dye et al., 1995; Foley et al., 2011; Hayden et al., 2011; Stanier et al., 2021). The shoreline communities of Lake Michigan have historically been in non-attainment of federal ozone standards. Precursors to ozone production have emission sources along the Chicago urban corridor and ozone production can be enhanced when those emissions are trapped in the marine layer over the lake and get transported northward from Chicago (Vermeuel et al., 2019; Dye et al., 1995; Foley et al., 2011). The low altitude features in ozone gradients over Lake Michigan have been observed in the recent 2017 Lake Michigan Ozone Study field campaign (Stanier et al., 2021; Doak et al., 2021). Stanier, et al. (2021) describe observations for the highest measured ozone during the field campaign existing over water, offshore from Milwaukee and in the altitude range of 30-100 m above lake level. The shallow marine layer disruption when crossing a shoreline boundary during a lake breeze is a unique environment to study the changes in vertical mixing and pollutant extent.

WiscoDISCO-21 featured round-based Doppler lidar observations and two Uncrewed Aircraft Systems (UAS), including the University of Colorado RAAVEN fixed-wing UAS and Purdue University's DJI M210 quadcopter. These platforms were deployed to enhance the level of detail and extend the domains of ground-based measurements to manned aircraft observations with higher spatial resolution and sustained lower-altitude flight. UAS are well suited to make observations of a shoreline environment impacted by a shallow marine layer, where vertical mixing and pollutant transport are key to understanding pollution events at the surface. UAS have been used in riverine environments to highlight pollutant transport and night time boundary layer dynamics (Guimaras et al., 2020). During the Ozone



Water-Land Environmental Transition Study (OWLETs), UAS, ozone sondes and lidar observations were
used to observe ozone titration events above the Chesapeake Bay shipping channel (Gronoff et al., 2019).
Horel et al., (Horel et al., 2016) describe the use of distributed ground sensors, tethered sondes and UAS
to better understand the meteorological and pollutant transport factors surrounding poor air quality in the
Salt Lake Valley. Incorporation of multi-hole probes into fixed-wing UAS has allowed for observations
of 3-D winds (Elston et al., 2015) and turbulent fluxes (Wildmann et al., 2014). The RAAVEN platform
leveraged in WiscoDISCO-21 has recently been used to study the lower atmosphere across a variety of
environmental regimes. This includes nearly a month of flight operations to investigate the
thermodynamic and kinematic structure of trade winds over the tropical Atlantic Ocean, (de Boer et al.
2021a) as well as deployments to the US Midwest to make measurements of supercell thunderstorms
(Frew et al., 2020). The measurement accuracy of the RAAVEN's instrumentation was recently evaluated
at the US Department of Energy's Atmospheric Radiation Measurement (ARM) program's Southern
Great Plains facility (see de Boer et al. (2021b)  for details).

Such high-resolution UAS observations are well-suited for documenting and characterizing the
dimensions of the lake breeze phenomenon and corresponding pollutant transport. A combination of UAS
and lidar can provide overlapping domains of observations that scale up to planetary boundary layer
heights, with  UAS providing detailed insight into nonuniformities in meteorological observations along
the Lake Michigan shoreline. UAS observations are particularly complementary to Doppler lidar
observations, as such observations are subject to near-field issues that prevent them from making
observations at very low altitudes. Given that the UAS readily operate between the surface and 100 m,
these platforms can fill in this gap and provide detailed insight into thermodynamic, kinematic and
chemical properties of this layer. These observations have higher vertical and temporal resolution than
many chemical models which can provide insight into model resolution issues at the lake-land interface
(Wagner et al., 2021). The WiscoDISCO-21 field campaign was conducted in conjunction with the
Enhanced Ozone Monitoring initiative from the Wi-DNR who housed added instrumentation for NO,
$NO_y$, $NO_x$, VOC canisters and PANDORA instrumentation at the Chiwaukee Prairie air monitoring





station. The WiDNR has provided a portal for access to data from these sensors through their web portal
(https://wi-dnr.widencollective.com/portals/iwvftorq/AirMonitoringData).

**2. Description of measurement location, deployment strategies and sampling**
The Chiwaukee Prairie State Natural Area is a 485-acre shoreline prairie managed by the Wisconsin
Department of Natural Resources (WiDNR) located along the shoreline of Lake Michigan and adjacent
to the Wisconsin/Illinois border. The WiDNR operates an air monitoring station (Airs ID 55-059-0019)
for Kenosha County within this area, located at 11838 First Court in Pleasant Prairie, WI. This location
was chosen due to its suitability for UAS flight operations and the regular influence of lake breeze
circulations at the site. As a result of these lake breezes, the WiDNR's Chiwaukee Prairie Monitor
regularly observes some of the highest ozone concentrations in the state (Stanier et al., 2021). Land use
in the region is mixed suburban housing and farming, with two marinas directly south of the research site.
Chiwaukee Prairie has trail access along Al Kemper Trail and 122nd Street that is isolated from
automobile, bicycle and most pedestrian traffic. The M210 flights were conducted near to the WiDNR
Air Monitoring site (Latitude: 42.5045, Longitude: -87.8095) and the RAAVEN flight operations were
conducted on Al Kemper Trail or 122nd St to provide ample room for take-off and landing (Fig. 1).





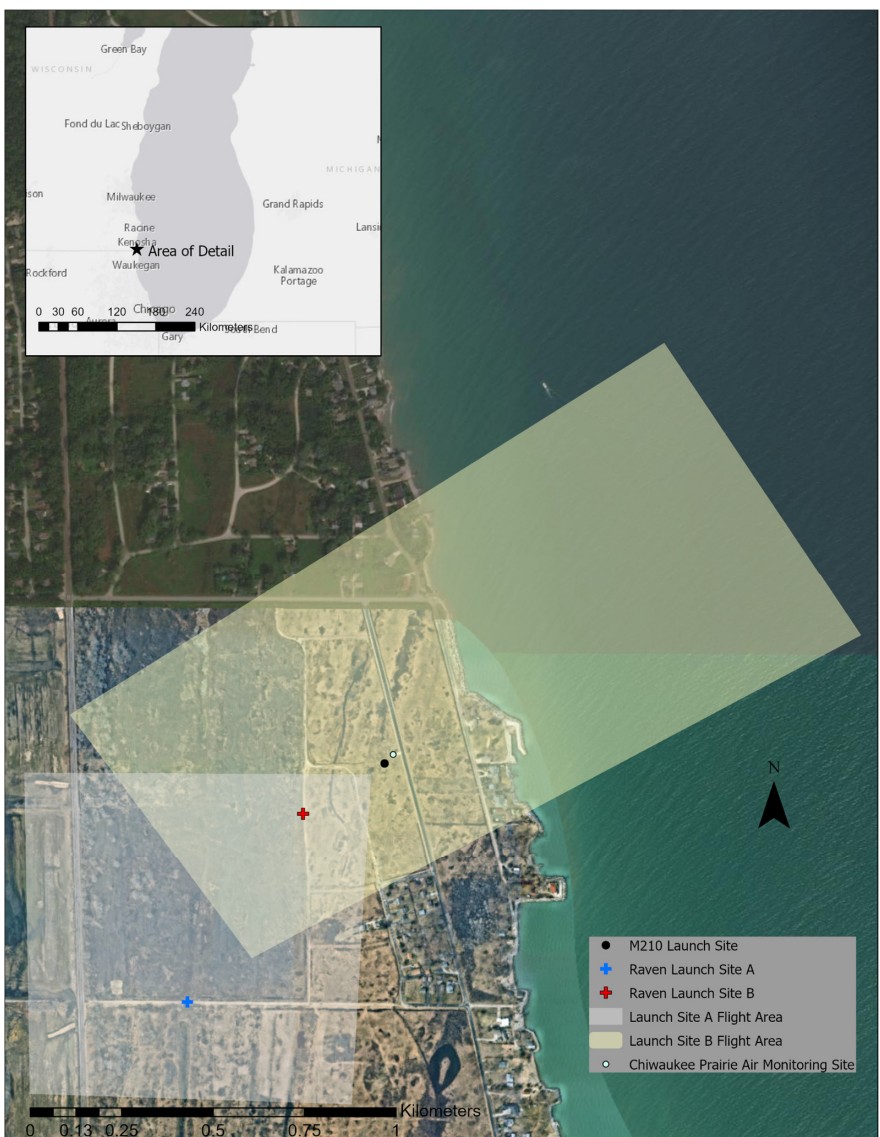


**Figure 1**: Research site map including Chiwaukee Prairie air monitor and locations for launch sites for
M210 and RAAVEN. Map created using Esri ArcPro version 2.52 using ArcPro basemap imagery.







The primary goal for the field campaign was to capture elevated ozone concentration events resulting
from lake breeze circulations at the site. The deployment strategy for selecting a time window for field
operations was dictated by ozone and meteorological forecasts that predicted light southerly winds for an
extended period that would both a) increase the likelihood of onshore lake breeze flow from weaker
southerly winds and b) drive pollutant transport from the Chicago metro area to the site. Forecasts from
both the WiDNR and Realtime Air Quality Modeling System (RAQMS) were used to select an ideal
deployment period. The dates of May 21-26, 2021 were chosen as meeting those requirements. Flights
were cancelled during days in which high ozone or southerly/southeasterly lake breeze were not expected
(Table 1).

**Table 1:** UAS flight days and conditions for the WiscoDISCO-21 field campaign

| Date    (2021) | M210 (time UTC) | University of Colorado RAAVEN (Time UTC and flight pattern) | Weather and Air Quality Conditions |
|---|---|---|---|
| Friday, May 21 | F1 (15:35-15:44)<br>F2 (16:38-16:47)<br>F3 (19:08-19:21)<br>F4 (19:46-19:59) | F1 (15:01-16:54)<br>Pattern A<br>F2 (18:36-20:40)<br>Pattern A | SW wind, Temps > 25 $^o$C, small shift in winds to colder from SSE |
| Saturday, May 22 | F1 (14:22-14:35)<br>F2 (15:18-15:31)<br>F3 (17:27-17:41)<br>F4 (18:26-18:41)<br>F5 (20:09-20:22)<br>F6 (20:59-21:14) | F1 (13:52-15:55)<br>Pattern A<br>F2 (17:00-19:03)<br>Pattern A<br>F3 (19:30-21:38)<br>Pattern A | W wind in AM, Temps > 25 $^o$C, consistent shift in winds to colder from SSE. |
| Sunday, May 23 | no flights | no flights | W to NE winds, dropping temperatures, AM showers, PM showers |
| Monday, May 24 | F1 (15:08-15:23)<br>F2 (16:01-16:16)<br>F3 (18:14-18:29)<br>F4 (19:12-19:27) | F1 (14:24-16:30)<br>Pattern B<br>F2 (17:41-19:50)<br>Pattern B | S winds, lake breeze, high ozone event (> 70 ppb). |



| | F5 (21:09-21:19)<br>F6 (22:04-22:19) | F3 (20:42-22:51)<br>Pattern B | |
|---|---|---|---|
| Tuesday, May 25 | F1 (14:00-14:15)<br>F2 (14:49-15:04) | F1 (13:39-15:42)<br>Pattern B | SW winds, slight lake breeze in the morning, overtaken by westerlies |
| Wednesday, May 26 | F1 (13:43-13:58)<br>F2 (14:37-14:52)<br>F3 (16:47-17:02)<br>F4 (17:47-18:01)<br>F5 (19:51-20:06)<br>F6 (20:48-21:01) | F1 (13:27-15:24)<br>Pattern B<br>F2 (16:31-18:20)<br>Pattern B<br>F3 (19:30-21:22)<br>Pattern B | W wind, steady all day, sunny. After all flights, lake breeze came in from NE |


Flights were conducted in the time window 08:00-17:00 local time, CDT (13:00-22:00 UTC) (Table 1).
The RAAVEN platform features 2-hour flight times and was deployed to complete up to 3 flights per
day. The M210 flew slow ascents to 120 m AGL with an approximate 15-minute flight time, completing
up to 6 flights per day and the sampling pattern was designated to achieve maximum overlap with the
RAAVEN flight times by conducting two flights per RAAVEN flight.

During WiscoDISCO-21, the RAAVEN completed 12 flights, totaling 25.4 flight hours, operating under
a Certificate of Authorization (COA) from the US Federal Aviation Administration (FAA) to allow flights
up to 518 m AGL.  Fig. 2a shows a map of the RAAVEN flights, while figure 2b includes a histogram of
the altitudes covered by these flights.  Flights were designed to follow two distinct flight patterns to
capture over-prairie profiles using a circular pattern with holding at altitudes 400, 250, 150, 100 and 50
m AGL and over-water/over-prairie profiles using an extended racetrack pattern that traversed the
shoreline, with holding altitudes at 400, 250, 150, 100 and 50 m AGL (see Figure 2c for the two flight
patterns).  Figure 3 shows histograms of the measurements obtained by the RAAVEN over the length of
the campaign, including temperature, relative humidity, wind speed, wind direction, air pressure, and
surface and sky brightness temperature.

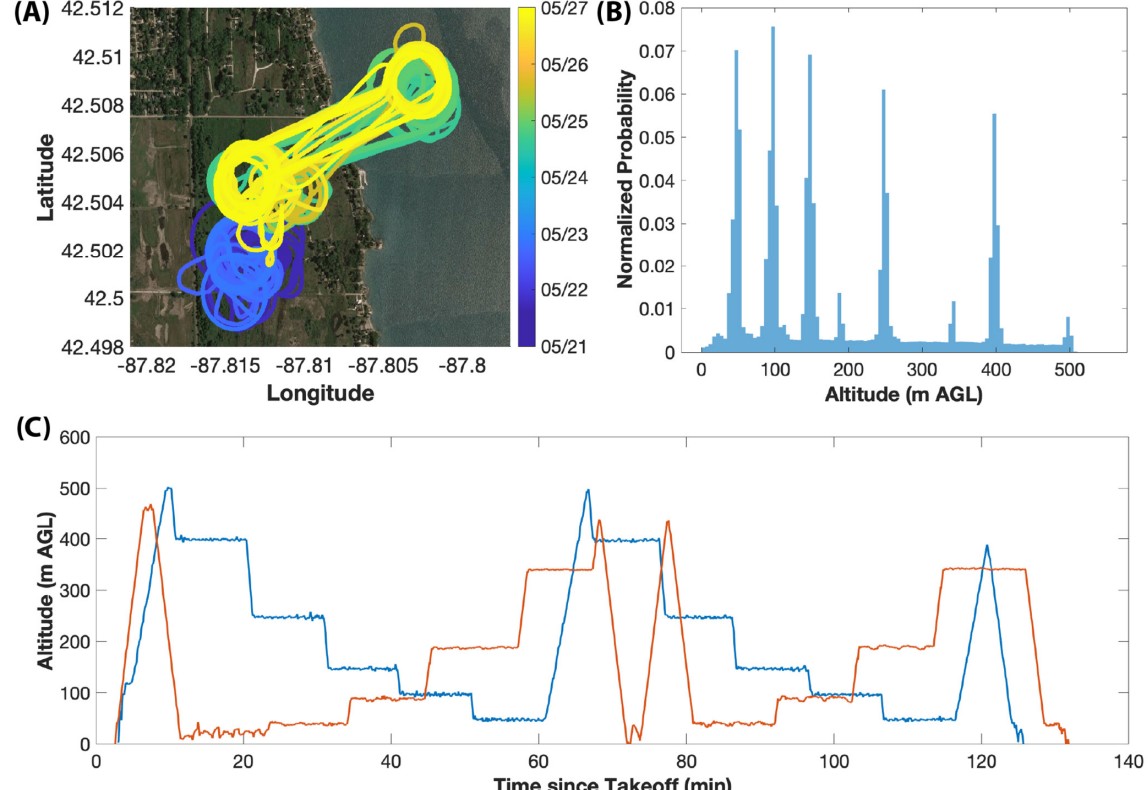

**Figure 2:** A map showing the flight tracks for the RAAVEN (a), a histogram of altitudes sampled by the
RAAVEN (b), and example time-height plots of the two types of RAAVEN flights (c). Background maps
are © GoogleMaps 2021, downloaded through their API.

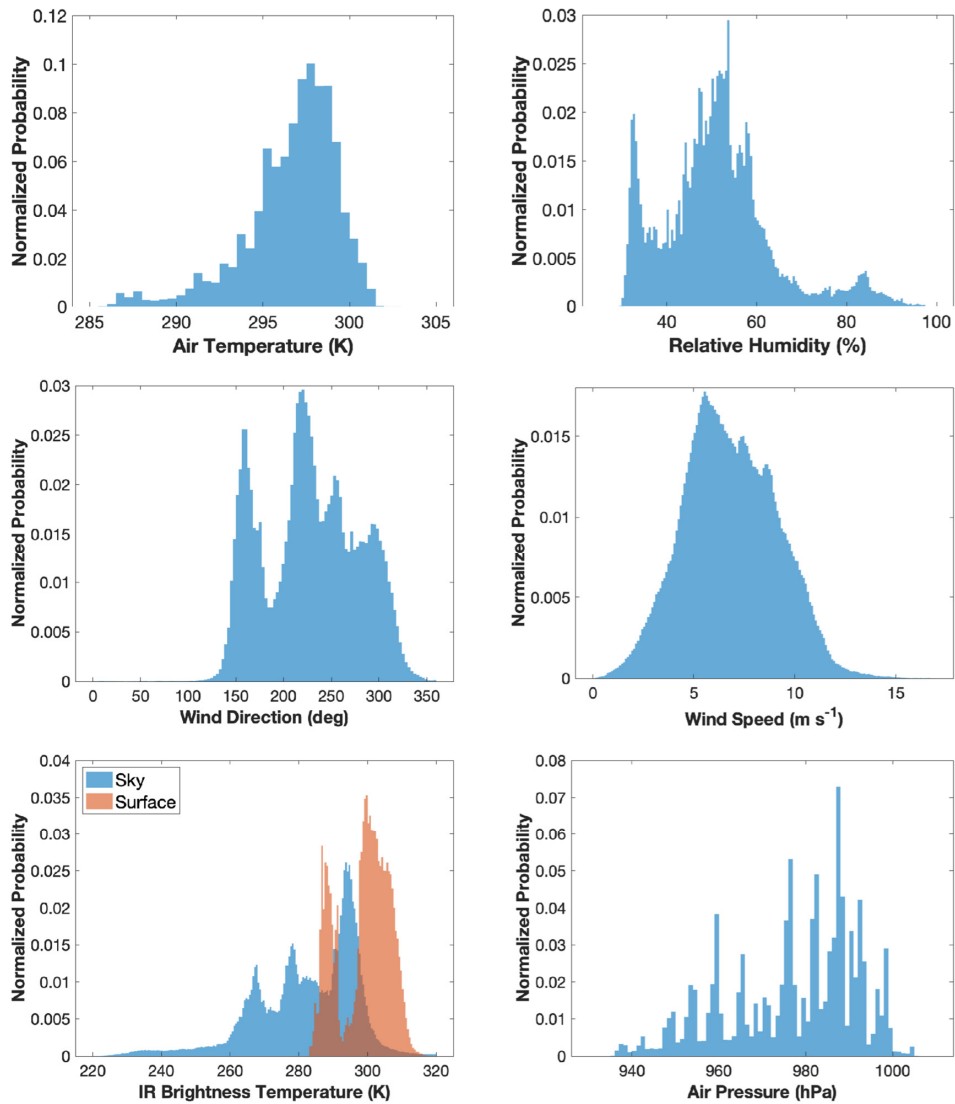

**Figure 3:** Histograms of (clockwise from top left) temperature, relative humidity, wind speed,
pressure, IR brightness temperatures, and wind direction, as measured by the RAAVEN during
WiscoDISCO-21.





**3. Description of Instrumentation and Vehicles**
**3.1 University of Colorado RAAVEN UAS**
The RAAVEN UAS (Fig. 4) is a fixed-wing UAS with a wingspan of 2.3 m that has been operated by the
University of Colorado Boulder since 2019. The RAAVEN's airframe is based on a custom-manufactured
model from RiteWing RC. The airframe has been updated to meet the needs of atmospheric science
missions spanning a variety of environments. The RAAVEN leverages the PixHawk2 autopilot system
and employs an 8S 21000 mAh Lithium Ion (Li-Ion) battery pack to offer flight times around 2.5 hours,
depending on conditions and executed flight patterns.  Specific modifications to the airframe include the
integration of a tail boom to enhance longitudinal stability and improve the platform's performance. The
aircraft has a top airspeed of approximately 130 km hr , though operations during WiscoDISCO-21 were
almost exclusively conducted in the 60-70 km hr$^{-1}$ range.

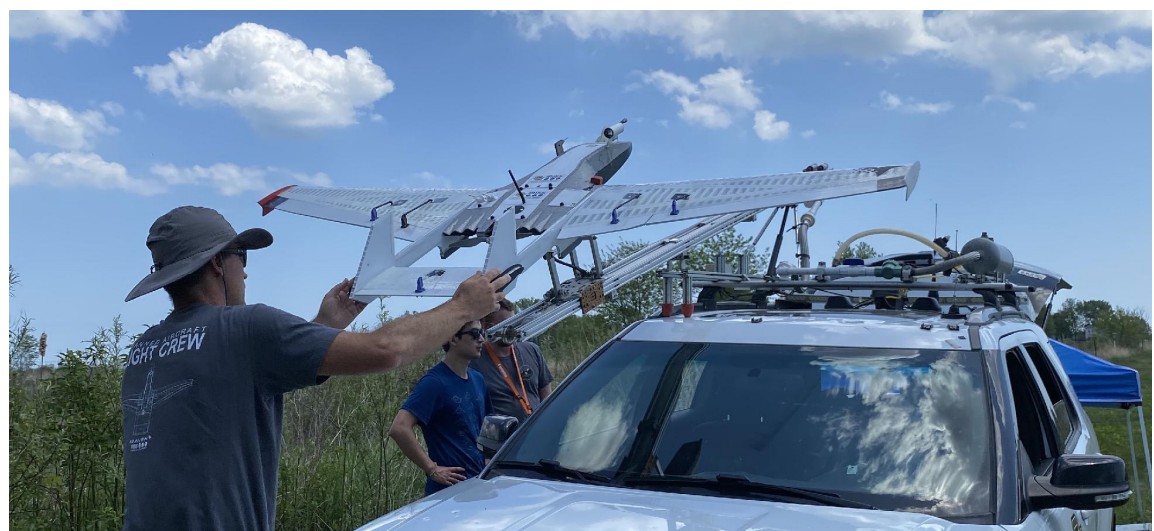





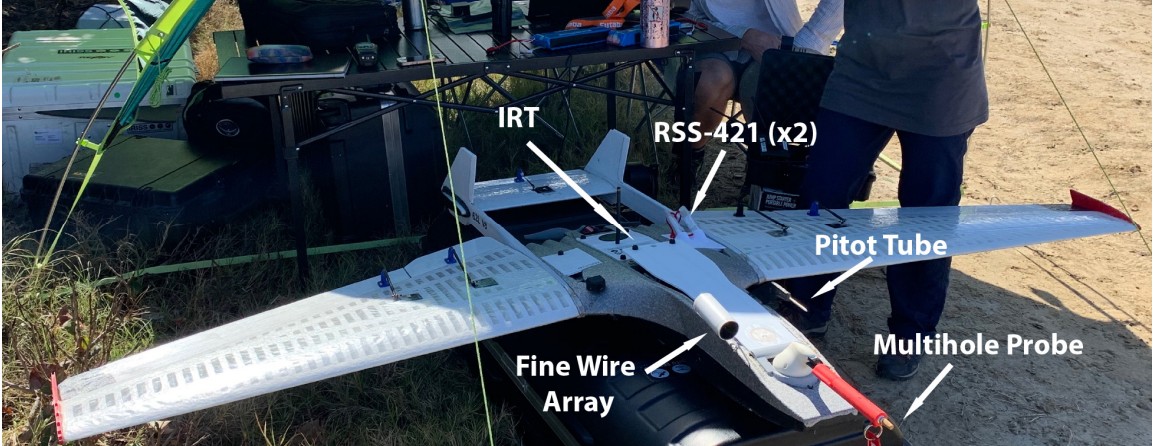

**Figure 4:** The University of Colorado RAAVEN being prepared for launch during WiscoDISCO21
(top), and a close up of the RAAVEN sensing systems (bottom).


For the WiscoDISCO-21 campaign, the RAAVEN was equipped with an instrument suite derived from
the *miniFlux* payload co-developed by the National Oceanic and Atmospheric Administration (NOAA),
the Cooperative Institute for Research in Environmental Sciences (CIRES) and Integrated Remote and In
Situ Sensing (IRISS) at the University of Colorado. In this configuration, the aircraft was set up to
measure atmospheric and surface properties to support evaluation of thermodynamic state, kinematic
state, and turbulent fluxes of heat and momentum. This involves a suite of core instrumentation (see Fig.
3), including a multihole pressure probe (MHP) from Black Swift Technologies, LLC (BST), a pair of
RSS421 PTH (pressure, temperature, humidity) sensors from Vaisala, Inc., a custom finewire array,
developed and manufactured at the University of Colorado Boulder, a pair of Melexis MLX90614 IR
thermometers, and a VectorNav VN-300 inertial navigation system (INS). This sensor suite is logged
using a custom- designed FlexLogger data logging system.

The Vaisala RSS421 sensors are identical to those used in the Vaisala RD41 dropsonde, and very similar
to the Vaisala RS41 radiosonde. This unit employs a linear resistive platinum temperature sensor with a
resolution of 0.01 °C, repeatability of 0.1 °C and a response time (as measured within the RS41





radiosonde) of 0.5 s at 1000 hPa when moving at 6 m s⁻¹. For relative humidity (RH), the RSS421 includes
a thin-film capacitor with a resolution of 0.1% RH and a repeatability of 2% RH, with a temperature-
dependent response time of better than 0.3 s at 20 °C (again, as measured within the RS41, with 6 m s⁻¹
airflow at 1000 hPa). Finally, the pressure sensor is capacitive with a silicon diaphragm, having a
resolution of 0.01 hPa and a repeatability of 0.4 hPa. For WiscoDISCO-21, a pair of these sensor modules
was mounted to the top of the RAAVEN's fuselage, between the nose and the tail of the aircraft on the
port side. The sensor mounting angles were offset to ensure that the two sensors would have different
amounts of solar exposure as the aircraft maneuvers through the atmosphere and to allow for the detection
of solar heating effects since no shading is used. Additional information on atmospheric thermodynamic
state is available from an E+E EE-03 sensor that is integrated into the BST MHP and from a Sensiron
SHT-85 sensor that is integrated in the custom finewire array. The EE-03 has a temperature accuracy (at
20 °C) of 0.3 °C, while the humidity accuracy is stated to be 3% RH at 21 °C. The SHT-85 has a stated
temperature accuracy of 0.1 °C (from 20-50 °C) and a repeatability of 0.08 C, while the humidity sensor
has a stated accuracy of 1.5% RH and a repeatability of 0.15 % RH. Both the EE03 and the SHT-85
sensors have slower response times than the RSS421 sensor described above and are typically not used
for scientific purposes unless there is a complete failure of the RSS421.

In addition to the SHT-85 sensor, the finewire array consists of two 5 μm diameter platinum wires
extending over a 2 mm length, suspended in the free stream by supporting prongs. One wire is operated
as a hotwire anemometer, with approximately 100 °C overheating compared to the ambient environmental
temperature. The other wire is operated as a coldwire thermometer, with approximately 1 °C overheating
relative to the surrounding environment. The wires have thermal time constants of 0.5 ms in a 15 m s⁻¹
airflow regime and support a sampling frequency of up to 800Hz to support measurement of turbulent
fluctuations in velocity and temperature. An electronics module converts resistance change in the wires
due to velocity or temperature variability to amplified voltages. For WiscoDISCO-21, the finewire was
logged at 250 Hz by the FlexLogger, which is equivalent to a 7.2 cm minimum length scale at the
RAAVEN's typical cruise airspeed of 18 m s. Time series of these recorded data are processed during
post-flight analysis to transform the voltages recorded by the fine wire module to velocity and





temperature. Additionally, these measured quantities can be fit to inertial sub-range turbulence models to
wavenumber spectra over suitable time intervals, producing turbulence intensity parameters epsilon
(kinetic energy dissipation rate) and $CT^2$ (temperature structure constant). The resolution (noise floor) of
these parameterizations is $2.0 \times 10^{-7}$ W kg$^{-1}$ for epsilon and $4.5 \times 10$ K m for $CT^2$. Resolution of the raw time
series are $8.3 \times 10$ m s for the hotwire and $1.3 \times 10$ K for the coldwire.

In addition to the EE-03 PTH measurements, the BST 5-hole probe supports measurement of airspeed,
angle of attack ($\alpha$) and sideslip angle ($\beta$). These measurements are combined with the GPS-based ground
velocities and aircraft attitude from the VectorNav VN-300 to derive the three-components of the inertial
wind (u, v, w), as discussed in section 4. The VN-300 can be configured in a dual-Global Navigation
Satellite System (GNSS) mode, under which the relative positions of two GNSS antennae are used to
calculate the platform yaw. However, this setting was not used during the WiscoDISCO-21 deployment.
Under dynamic conditions, the system has a stated accuracy of 0.3 degrees in GPS- Compass heading,
0.1 degrees in pitch and roll, 2.5 m horizontal position accuracy, 2.5 m vertical position accuracy when
integrating information from the barometric pressure sensor, and 0.05 m s$^{-1}$ accuracy in inertial velocity.
Input from the system's gyroscope, accelerometer, GNSS receiver, magnetometer and pressure sensor are
filtered through an extended Kalman Filter (EKF) to produce a navigation solution. VN-300 data were
logged at 50 Hz resolution during WiscoDISCO-21.

Finally, RAAVEN deploys two Melexis MLX90614 IR thermometers: one looking up from the top of
the aircraft and one looking down towards the surface in level flight. These sensors are factory calibrated
to work in operational temperatures between -40 and 125 C, and to measure target brightness temperatures
between -70 and 380 °C. They have a high accuracy (0.5 °C) and a measurement resolution of 0.02 °C.
The RAAVEN-mounted MLX90614s are not stabilized to maintain a vertical orientation, meaning that
the observed target is perpendicular to the reference frame of the aircraft. This requires some care when
interpreting measurement from time periods when the aircraft is conducting pitch or rolling maneuvers.
For WiscoDISCO-21, we leveraged the "I" version of this sensor which has a 5-degree field of view.
These sensors have a broad passband range of 5-14 µm, meaning that while it covers the infrared





atmospheric window, it is also subject to radiation emitted by water vapor and other radiatively active
gases, meaning that a significant depth of atmosphere between the aircraft and a given target (e.g., cloud;
surface), atmospheric gases influence the temperature reading. Despite this range spanning the 9.6 μm $O_3$
band, the relative proximity of the sensor to targets of interest (e.g. surface, clouds) means that this overlap
is not expected to significantly influence the readings, due to the integrated path length being relatively
small. Therefore, if absolute accuracy of brightness temperature is important, the sensor should be
operated in close proximity to a target of interest. However, relative contributions from different surface
types or atmospheric conditions can still be easily distinguished despite a lack of absolute calibration for
extended distance sensing. Such gradient detection can be useful for detecting surface inhomogeneities,
or for understanding whether the aircraft is operating under cloud or clear sky.

**3.2 M210 UAS**
The DJI M210 quad copter was equipped with a 3-D printed top-mounted bracket for holding a 2B
Technologies Personal Ozone Monitor (POM) and an Intermet Systems iMET-XQ2 meteorology sensor
(Fig. 5). The copter had a ~15 minute flight time with the on-board sensors without a camera. The POM
measures ambient ozone using UV absorption and active humidity subtraction by measuring a whole air
sample and an ozone scrubbed sample in a 10-s duty cycle. The POM records data to its internal data
storage at 10 s interval with a log number and time stamp along with GPS coordinates and instrumentation
metrics (optical cell pressure and temperature). The iMET system records temperature, humidity, and
pressure along with GPS coordinates and a time stamp to internal data storage. Each instrument (the POM
and iMET) had individual data logging systems and separate power supplies. Both the POM and the iMET
had GPS capabilities with the POM logging inconsistently and the iMET logging GPS more consistently.
Each instrument and the UAS flight recorder logged time stamps. The iMET recorded observations of
temperature, relative humidity, humidity temperature and pressure at a frequency of 10 Hz. The POM
recorded ozone observations at a frequency of 0.1 Hz. The POM, iMET and M210 time-stamps drifted
up to 60 seconds from the other logged data. The flight log recorded the M210 positioning (altitude,
latitude, longitude) at 100 Hz. The M210 flight logs, iMET data and POM data were each downloaded
separately after each series of flights.

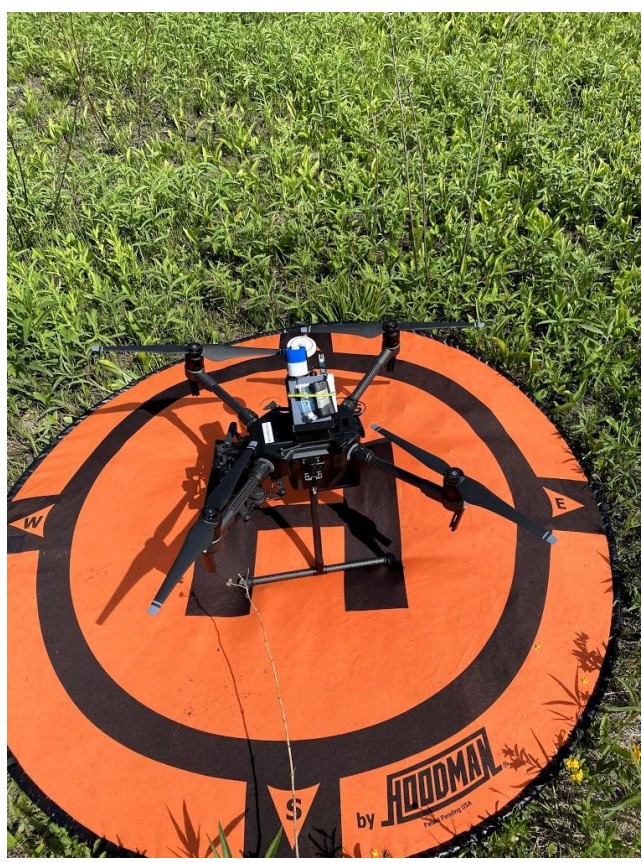

**Figure 5**: DJI M210 multirotor UAS with bracket-mounted POM and iMET

**3.3 Chiwaukee Lidar System**
A Halo Photonics Stream Line XR Doppler lidar (Pearson et al. 2009) was deployed on the roof of the
Chiwaukee Prairie air monitoring station (Fig. 6), approximately 3 m AGL. This is the same system that
is regularly deployed as part of the Space Science and Engineering Center (SSEC) Portable Atmospheric
Research Center, SPARC, (Wagner et al., 2019). The Doppler lidar actively emits pulses of near-infrared



radiation at a wavelength of 1.5 μm. This wavelength is long enough that molecular scattering causes
little attenuation of the signal, but it is short enough that it is sensitive to aerosols that are suspended
within the planetary boundary layer.

The Doppler lidar uses velocity-azimuth display (VAD) scans of the Doppler lidar to retrieve profiles of
wind speed and direction. In VAD, an instrument capable of measuring along-beam velocity (like a
Doppler radar or lidar) stares at multiple azimuths at a non-zenith elevation angle over a short period of
time, and then reconstructs the profile of winds above the lidar by assessing how the along-beam velocity
changes as a function of azimuth and range. For WiscoDISCO-21, the VAD scans were configured with
six azimuthal stares per profile (at azimuths of 0 º, 60 º, 120 º, and so on) with an elevation angle of 70 º.
Range gates were 18 m. VAD scans were conducted every 4 min and each VAD took approximately 45
s to complete. Between VADs, the lidar reverted to vertical stares in order to capture profiles of
backscatter and vertical velocity. Since the lidar depends on the presence of scatterers to have a detectable
signal return, the depth of the retrieved wind profiles varied significantly throughout the experiment from
as shallow as 200 m to as deep as 2 km.

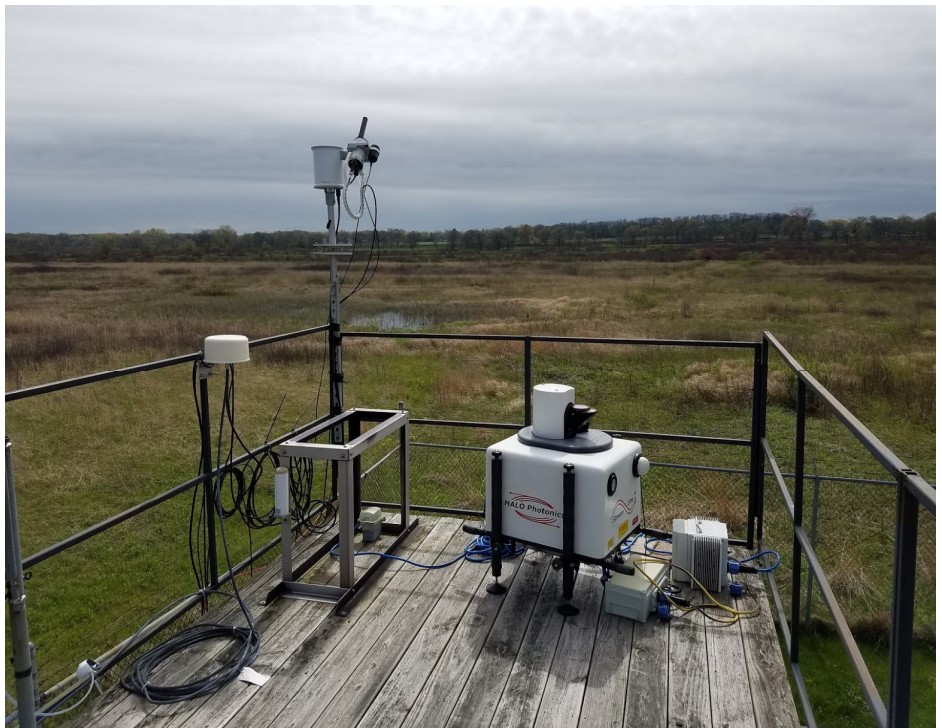

**Figure 6:** Roof of the Chiwaukee Prairie air monitoring system, showing the PANDORA (upper left)
and Doppler lidar (right-center). The wooden floor pictured here is approximately 3 m above ground
level.

**4. Data processing and quality control**
**4.1 University of Colorado RAAVEN UAS**
Data collected by the different sensors carried by the RAAVEN during WiscoDISCO-21 were logged at
a variety of different logging rates. The finewire system was logged at 250 Hz, the fastest rate of all of
the sensors. The BST MHP was logged at 100 Hz, the VectorNav VN-300 at 50 Hz, the Melexis IR
sensors and variables related to finewire status were logged at 20 Hz, while data collected from the
PixHawk autopilot and Vaisala RSS421 sensors were logged at 5 Hz. Each logging event carried out by
the FlexLogger includes a sample time from the logger CPU clock, allowing for post-collection time



alignment between the different sensors. These sample times, along with artificial 5, 20, 50, 100, and 250
Hz clocks spanning the sample times between initial GPS lock and the last recorded sample time for the
VN-300, are used to align the different variables to a set of common clocks, primarily through one-
dimensional linear interpolation. One exception to the linear interpolation is the yaw estimate, which is
circular in nature (ranging between -180 and 180 degrees), and therefore uses a "nearest" interpolation to
ensure that the transition from 360 to 0 degrees is not represented as 180. During this interpolation
process, a limited number of points sharing a common sample time with another point are removed from
the record. Once these time variables are established, a *base_time* variable is established using the 250
Hz time stamp, and offsets from *base_time* are then calculated for all different logging resolutions.

The resampled (in time) dataset includes a variety of derived and measured quantities. Aircraft position,
including latitude, longitude, and altitude, are measured by the VN-300. The aircraft altitude is corrected
using a combination of various inputs from onboard GPS and pressure altimeters, as neither of these
altitude estimates can be used reliably as a definite flight altitude. Pressure altitude is subject to drift over
the duration of a single flight due to atmospheric evolution over a 2.5-hour window, potentially resulting
in values at landing that are higher or lower than those at take-off. Similarly, the accuracy of the GPS
altitude is insufficient to capture the vertical position of the aircraft to the level of detail required. To
calculate a true altitude, a combination of the autopilot altitude, VN-300 altitude, and VN-300 pressure
are used. First, a *flight_flag* variable is computed using airspeed and altitude information from the
autopilot. Any data points with airspeed exceeding 10 m s$^{-1}$ and an altitude exceeding 5 m AGL is flagged
as a time when the aircraft is flying (*flight_flag*=1). The point at 200 samples (4 seconds) prior to the
first point in the record where the aircraft is deemed to be flying is recorded as the initial take-off index,
while the data point at 200 samples (4 seconds) after the last point in the record where the aircraft is
deemed to be flying is recorded as the landing index. The difference between the autopilot altitude at
these two indices is added into the flight record on a timestep-by-timestep basis, to correct for temporal
drift in pressure. A linear fit is then calculated to relate the VN-300 pressure and the difference between
the VN-300 reported altitude and the autopilot reported altitude. This pressure-dependent altitude
correction is then applied to the VN-300-reported altitude to derive a final altitude.




Wind estimation from fixed-wing aircraft requires the combination of several different measurements
related to airspeed, aircraft motion, and airflow over the aircraft (see van den Kroonenberg et al., 2008).
These measurements need to be of sufficient quality, and angular offsets and logging delays need to be
considered and removed.  For RAAVEN, true airspeed (TAS) biases have a large impact on derivation of
wind speed, while the angular offsets between the MHP and INS and time-lag between the GPS and in
situ measurements have smaller impacts.  These potential sources of error are corrected for using an
optimization technique, where small adjustments are made to the individual parameters and the
combination that results in the wind solution with the smallest overall variance is selected as the correct
winds.

For the RAAVEN WiscoDISCO-21 dataset, TAS is calculated using measurements from the MHP and
RSS421 probe using equation 1 from (Brown et al., 1983):

$$TAS_i = \sqrt{\frac{2\bar{q}}{\rho}} \tag{1}$$

where $\rho$ is the density of air calculated using the static pressure reported from the MHP, temperature from
the RSS421, and the specific gas constant for dry air, $\bar{q}$ is defined as:

$$\bar{q} = \frac{p_0}{1 - \frac{9}{4}sin^2\,\theta_a} \tag{2}$$

where $sin^2\,\theta_a$ is the total aerodynamic angle of the MHP, calculated using the angle of attack ($\alpha$) and
sideslip angle ($\beta$) reported by the MHP.

Based on testing in a temperature chamber, the pressure sensors used in this version of the MHP were
found to have non-linear temperature dependencies.  This requires the application of an additional
temperature-dependent correction to ensure that an artificial alteration of TAS with altitude was not
present.  Additional information on the calculation of airspeed and other quantities from the MHP can be
found in (de Boer et al., 2021a).



Derivation of the RAAVEN's thermodynamic measurements included multiple processing steps. First,
data from the two RSS421 sensors are averaged to attempt to reduce the influence of any solar exposure
of the sensors. Previous evaluations of the potential for solar contamination have not revealed any specific
biases on the observation (see de Boer et al., 2021a). Over the course of the WiscoDISCO-21 campaign,
the two sensors varied by less than 0.5 °C (Fig. 7). The averaged temperature time series was then used
to calibrate the coldwire data by applying a linear fit to the relationship between the coldwire voltage and
the temperature measured by the RSS421 sensor. The RSS421 relative humidity values were also
averaged. Typically, the RH measurements agreed to within 2%.

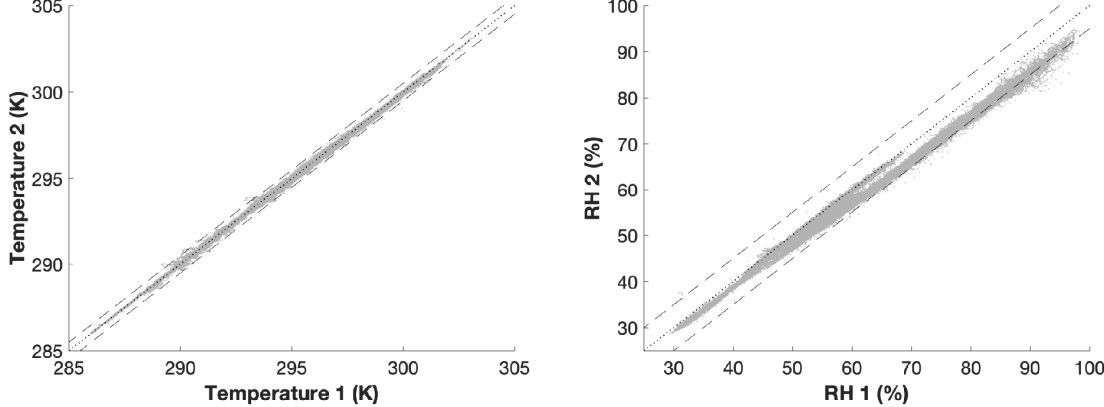


**Figure 7:** A comparison of temperature (left) and relative humidity (right) observations from the two
Vaisala RSS-421 sensors on RAAVEN for all flights. The dotted lines represent a one-to-one agreement,
with the dashed lines representing 0.5 degree (for temperature) and 5% (for relative humidity) deviation
from perfect agreement.


All quantities measured by the RAAVEN have data quality flags associated with them. For the RSS421-
derived temperature, the flag is set to zero for good data, and set to one for times when any of the following
occur: a) the absolute value of the difference between the temperature from either individual sensor and
the output temperature is greater than 0.5 °C, b) the absolute value of the difference between the output
temperature and the temperature from the EE-03 sensor on the MHP exceeds 5 °C, c) the recorded error



flag of either RSS421 sensor is active, or d) the aircraft is not flying. For the RH measurement from the
RSS421, a similar set of criteria are used to activate the data quality flag, except the limits are set to be
6% between RSS421 sensors, and 15% between the output RH value and the MHP-provided RH value.
The coldwire temperature data quality flag is activated when the difference between the coldwire
temperature and either of the RSS421 temperatures exceeds 1 °C, when the absolute value of the
difference between the coldwire temperature and the MHP temperature exceeds 3 °C, when the coldwire
voltage is observed to fall outside of the 0-4 V analog range, or when the aircraft is not flying. Finally,
the pressure quality control flag for the pressure measurement from the VN-300 is activated if the absolute
value of the difference between the reported VN-300 static pressure and that measured by either of the
RSS421 sensors exceeds 2.5 hPa. The RSS421 pressure measurements are not used because they are
believed to be biased low due to the airflow passing over their location on the aircraft.

In addition to the flags discussed above, we include a 3-stage flag for the wind measurements, which is
set to 0 (good data), 1 (suspect data) or 2 (bad data). Data are determined to be bad if any of the following
conditions were met:

-    The measured angle of attack or sideslip exceeds 20 degrees, with values between 10-20

degrees are flagged as "suspect"

-    The true airspeed (TAS) is below 10 m s$^{-1}$

-    Any of the MHP ports are deemed to be blocked, as determined by the differential pressure

value for any of the sensors falling below -100 Pa

-    The moving window variance of the MHP-derived TAS over 40 seconds is less than 5

-    The aircraft is not flying

-    The difference between the MHP TAS and that from the Pitot probe is greater than 5 m s$^{-1}$


Finally, we included two additional flags in the datastream to allow data users to better understand aircraft
flight state and support sampling during specific phases of flight. These flags include the "Flight_Flag"
introduced previously, as well as a "Flight_State" flag. The "Flight_State" flag includes information on
whether the is flying straight (0) or is turning (1) in the ones place, whether the aircraft is descending (0),





level (1), or ascending (2) in the tens place, and whether the aircraft is in flight (1) or not (0) in the
hundreds place. If, for example, a data user wanted to analyze straight, level flight legs, they would search
for data with "Flight_State" equal to 110. These flags are derived from information from a combination
of sensors, including the altitude variable described above, the aircraft yaw, and the "Flight_Flag" variable
described earlier on in this paragraph.

The accuracy of the RAAVEN observations has been evaluated in previous studies. For example, a
comparison of RAAVEN data with measurements collected by radiosondes launched from the Barbados
Cloud Observatory was conducted in recent work from de Boer et al., (2021b). While radiosondes in that
evaluation were launched approximately 20 km to the southeast, the air sampled by both systems was
largely representative of the marine boundary layer, implying limited spatial variability. In that
evaluation, the observations from the RAAVEN were very well correlated with those from the
radiosondes and do not show any positive or negative bias, supporting the idea that the RAAVEN
measurements provide an accurate depiction of the lower atmosphere. In addition, recent work allowed
for direct comparison of RAAVEN data to observations collected by radiosondes and a 60 m tower at the
US Department of Energy's Southern Great Plains (SGP) research site. That study, de Boer et al. (2021a),
similarly provided confidence in the RAAVEN observations, showing close statistical agreement between
the different data sources.

**4.2 M210 UAS**
Data from the M210 flight controller, the POM and the iMET were all logged to individual instrument
internal data storage with independent timestamps. The average flight time of the M210 was 13.96 min.
The POM instrument logged data every 10 s. The iMET logged data every 0.1 s and the M210 flight log
recorded UAS GPS positioning and flight statistics at 0.01 s intervals. The ozone concentrations from
the POM are adjusted to calibrated values, where ozone calibrations were conducted before every set of
2 flights for the M210 using a 2BTech Model 306 ozone calibration source (Fig. 8). Data quality flags
are established as 0 = no concern, 1 = time flag, 2 = calibration and time flag. The time flag indicates
flights where the time offset between the M210 and the instrument time offset is large (iMET > 10 s or



POM > 30 s). The calibration flag indicates when the POM was not responsive to the ozone calibration source (Flight 5 on May 24) after an over-water flight. All times were averaged to 90 seconds and compressed to the time window of observations for a single M210 ascent using the M210 timestamp. A time stamp for 90s averaged data from all instrumentation on the M210 was generated by using the M210 timestamp as primary and adjusting to a time offset in either the POM or the iMET for the start of a flight, then averaged each variable for every 90 second interval of the flight. A 1σ standard deviation is presented as the uncertainty for the 90-s averages. The iMET observations of temperature, relative humidity, pressure and humidity temperature are presented using the 90-s averages with uncertainty as 1σ standard deviations. Each flight ascent start and end were determined by observed changes in atmospheric pressure by the iMET sensor, altitude change in the M210, or noted time of ascent in field notebook for the POM. The altitudes for each observation were obtained by averaging the M210 flight log altitude data for the 90-s timestamps. The flight data timestamps varied slightly for each data source. The POM time drift was the most pronounced, with an average difference between the iMet of ~ 24 s. The POM's time was adjusted manually throughout the campaign as the time would drift over the course of one flight. The average difference between the iMet and the M210 over 20 flights was ~ 4 s. Only 20% of flights had a time-difference between iMET and M210 greater than 10 seconds. Instrument battery loss occurred for the iMET system which resulted in lost data for a few flights on May 26, 2021.



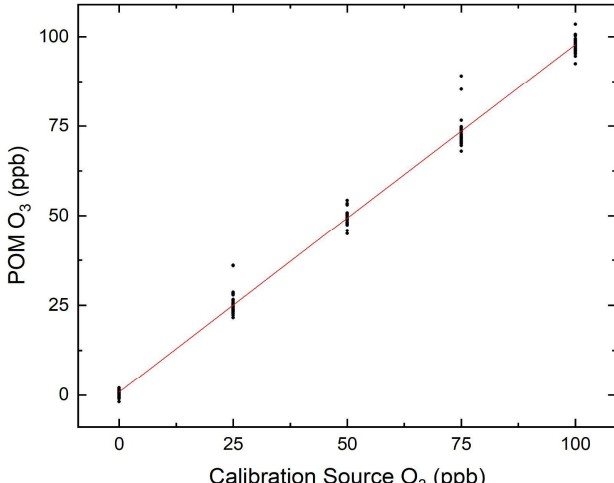

**Figure 8**: A sample POM calibration from May 24, 2021. The linear regression fit gives: $y = 0.9689$ (±
0.0061) $x + 0.83$ (±0.35), $R^2 = 0.9937$. Each calibration concentration had a 5-minute duration with the
POM logging 10-s data.

Intercomparison between observations made via instrumentation on the M210 at 5 m AGL and at the Wi-
DNR ground station show a linear agreement between the observations (Fig. 9). The linear agreement is
better for the iMET temperature and the ground station temperature with $R^2 = 0.970$ in comparison to $R^2$
= 0.955 for $O_3$ observations. The $O_3$ linear fit, $O_{3\,(POM)} = 0.944$ (± 0.044) $O_{3\,(DNR)}$ - 3.3 (±1.9), has a
negative intercept. The uncertainties in the POM's $O_3$ concentrations are much larger than uncertainties
in the ground station instrumentation. The linear agreement between the different instrumentation on
separate observation platforms demonstrates that the M210 platform instrumentation provides an
accurate, albeit less precise representation of the atmosphere.


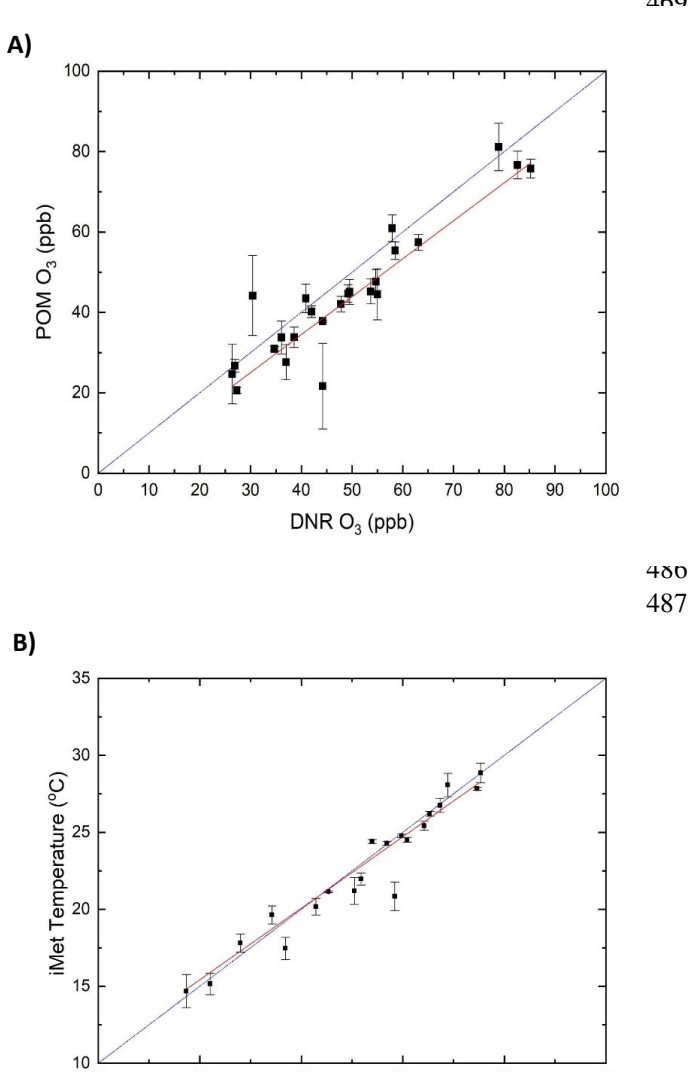

**Figure 9**: Intercomparsion between measurements from instrumentation on the M210 at 5 m AGL and at the WI-DNR ground station for a) $O_3$ (ppb) observations and b) temperature (° C). Blue lines depict 1:1 agreement and red lines depict the linear regression best fit with a) $O_{3\,(POM)} = 0.944\,(\pm\,0.044)\ O_{3\,(DNR)}$ - 3.3 ($\pm$1.9), $R^2 = 0.955$, and b) $T_{iMET} = 0.929(\pm\,0.038)\ T_{DNR} + 1.48(\pm0.93)$, $R^2 = 0.970$.





## 5. Data Availability and File Structure

A community data repository has been established for this field campaign at https://zenodo.org/communities/wiscodisco21/. The data sets in the repository cover the merged iMET and POM data sets from the M210 (DOI:10.5281/zenodo.5160346) as .txt files, the RAAVEN dataset (DOI: 10.5281/zenodo.5142491) as .cdf files, and the doppler lidar wind profiler (DOI: 10.5281/zenodo.5213039) as .cdf files. M210 files have a naming convention that includes WiscoDisco_M210_YYYYMMDD_F#, where the flight number for the day is indicated by F#. RAAVEN files have a naming convention that includes WiscoDisco_CU-RAAVEN_YYYYMMDD_hhmmss_B1.nc, where YYYYMMDD is the year, month and day that the data were collected, hhmmss is the time of power on for the aircraft, and B1 is the data processing level, where B1 files have had data quality checks and post-processing (e.g. coldwire calibration and wind estimation) applied. The Doppler lidar files have a naming convention that includes chiwaukee_wind_profiles_YYYYMMDD and chiwaukee_stare_YYYYMMDD. All datasets include geospatial information (latitude, longitude, altitude) and timestamps in UTC.

## 6. Summary

The 2021 WiscoDISCO field campaign incorporated the use of two UAS platforms for meteorological and chemical measurements in the atmosphere, a multirotor completing vertical profiles up to 120 m AGL and a fixed wing executing flight patterns up to 500 m AGL alongside a Lidar WindPro instrument capable of sensing winds and aerosol backscatter from altitudes of 100-2000 m AGL. The overlapping domains are useful for characterizing low altitude mesoscale meteorology of the lake breeze at a shoreline environment that regularly observes ozone enhancement events during onshore flow. Data from all instruments and platforms have been compiled, quality-control tested and uploaded to a community repository. The collaborative field campaign involved teams from 4 different universities and obtained continuous lidar data in conjunction with 24 flight hours of fixed wing and 6 flight hours of multi-rotor vertical profiles on days likely impacted by lake breeze.



The WiscoDISCO-21 project demonstrates how UAS can be used to sample a complex circulation
near to the surface without causing major disruption to people, wildlife and ecosystems in the area. An
example of a characterization of lake breeze incursion is shown in Figures 10 and 11, which include the
temperature profiles from the M210 and RAAVEN (Fig. 10) and Doppler lidar u wind component (Fig.
11). The temperature profiles from the M210 and RAAVEN show a notable temperature inversion in the
late afternoon below 150 m and the Doppler lidar u wind component shows easterly winds arriving after
18:00 UTC. The combination of u component winds from Doppler lidar and the temperature observations
from the UAS platforms are consistent in demonstrating a marine layer incursion with maximum height
of approximately 250 m AGL at 21:00 UCT collapsing to a height of 100 m AGL by 22:00 UTC. The
nonuniform start to the lake breeze onset fluctuated, shown as shifting u component winds from easterly
to westerly after 18:00 UTC (Fig 11) and disagreement with the lowest altitude observations from the
M210 and RAAVEN between 18:30-19:00 UTC (Fig 10). The distance between the M210 launch site
and the RAAVEN landing site complicates the low altitude observations of temperatures between 18:00
and 19:00 UTC, which also may indicate the very limited incursion of the lake breeze at that time.

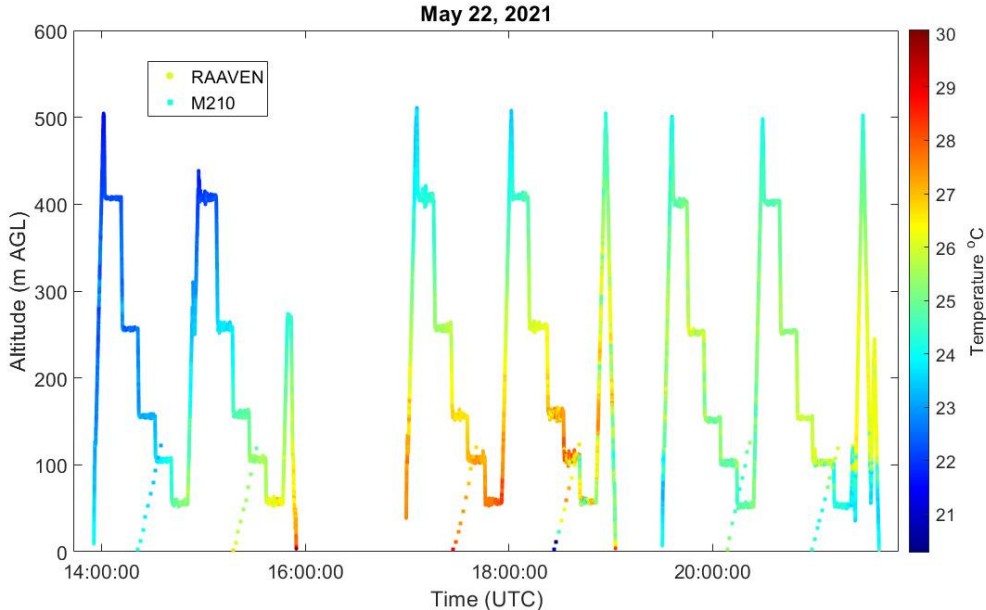




Earth System Open Access
Science Data Discussions


**Figure 10**: Temperatures (° C) measured from University of Colorado RAAVEN (O) and the M120 (□)
on May 22, 2021. RAAVEN was flying over-prairie circular spirals Pattern A.

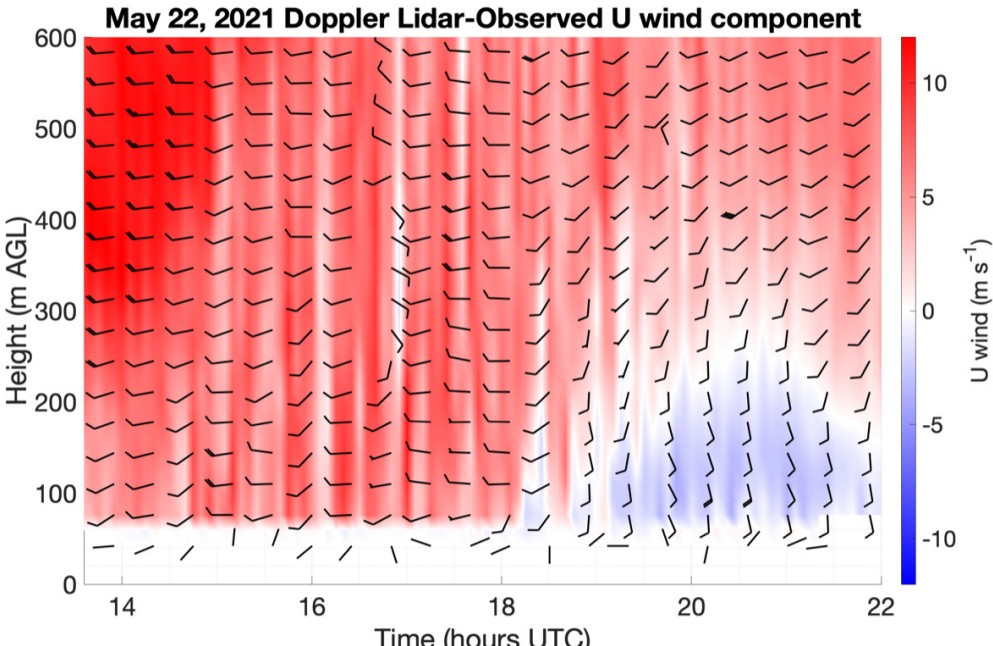


**Figure 11**: Time/height cross section of the u (zonal) component of the Doppler lidar-observed
horizontal winds (in m/s), overlaid with horizontal wind barbs (in kts) plotted according to the standard
convention from May 22, 2021.  Wind barbs are thinned by a factor of five in the time dimension and a
factor of two in the height dimension to aid readability.

The data from the WiscoDISCO-21 campaign can be used to evaluate the markers for lake breeze

incursion overland in winds, temperatures, chemical composition and optical properties (backscatter).
The thermodynamic conditions for lake breeze incursion at a local scale can be determined through the
evaluation of horizontal and vertical winds, atmospheric stability and potential temperature. The
positioning of pollutants with respect to the marine layer markers can also be investigated.
**Author Contributions.**



Patricia A. Cleary is the PI of this project and was responsible for data collection, overseeing data analysis
from the M210, field campaign planning and logistics, and the writing and editing of this document. Ben
Kies was responsible for data collection for the M210 in the field, Joe Tirado was responsible for data
analysis, quality control and data formatting for the repository for the M210, Aidan Voon was responsible
for data analysis for the M210. Joe Hupy was responsible for piloting the M210 and the writing and
editing of this paper. Gijs de Boer was responsible for coordination and execution of the University of
Colorado RAAVEN flights, and for development, writing and editing of the publication. Steve Borenstein
and Jonathan Hamilton contributed to the collection of the RAAVEN dataset as field operators, and
supported the development of this manuscript. Dale Lawrence supplied instrumentation for the RAAVEN
UAS and contributed to the writing of the manuscript. Tim Wagner and R Bradley Pierce were
responsible for data collection, data analysis of the doppler lidar instrumentation, and writing and editing
this document and R Bradley Pierce assisted in field planning.
**Competing Interests.**
GB works as a consultant for Black Swift Technologies, who manufacture the multi-hole pressure probe
used in the collection of the RAAVEN dataset.

**Acknowledgements**.
This material is based upon work funded by the National Science Foundation award #1918850. The UW-
Eau Claire team acknowledges support from the Student Blugold Commitment Differential Tuition
program. The University of Colorado team acknowledges financial support from the University of
Wisconsin Eau-Claire through a sub-contract supported by the US National Science Foundation, as well
as support from the NOAA Physical Sciences Laboratory. Any opinions, findings, and conclusions or
recommendations expressed in this material are those of the author(s) and do not necessarily reflect the
views of the National Science Foundation.






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
