# Peer review of "Observations of the Lower Atmosphere From the 2021 WiscoDISCO"

_Earth System Science Data, 2021_

## Referee Comment (RC1)

**Comments on "Observations of the Lower Atmosphere From 2021 WiscoDISCO Campaign" by Clearyet al. (2021)**

**Summary/recommendation:**

This paper focuses on conducting a field campaign "The 2021 WiscoDISCO" that aims at capturing the elevated ozone concentration events resulting from the lake breeze circulation at Chiwaukee Prairie State Natural Area in Southeastern Wisconsin from May 21-26 2021. To do that, they deployed two Unmanned Aircraft Systems (UAS) platforms at two different altitudes (500 m above AGL for University of Colorado RAAVEN fixed wing UAS and 120 m above AGL for Purdue University DJI multirotor UAS) to measure temperature, humidity, 3D wind Sand and vertical profiles of temperature, humidity and ozone. Moreover, the 2021WiscoDISCO was conducted concurrently with the Enhanced Ozone Monitoring plan from WI-DNR that included a Doppler lidar wind profiler deployed on the roof of the Chiwaukee Prairie air mentoring station.

The scientific approach is sound and the work presented is substantial. However, the paper deserves more work before publication. I do recommend the publication of the current manuscript in ESSD journal if the authors consider these minor revisions in order to make a nice addition to the literature. I request that the authors consider the following points as they revise this manuscript:

**General comments:**

1/ The introduction focuses mainly on the goal of the 2021WiscoDISCO campaign and needs more details to explain the main reason for conducting this campaign and highlight the importance of capturing the breeze impact on local observations by adding more references. Also, authors need to highlight if models capture the lake breeze impact on shoreline ozone observations based on previous studies.

2/ Authors should add a section to explain how Scientists from other fields could benefit from the database created during this campaign. The risk is that the impact of the publication on a broader scientific community remains limited unless the authors put the paper into a wider perspective. Thus, the authors should add a section/paragraph dedicated to a scientific discussion including more references to previous studies to highlight the importance of the collected data (other than determining the breezes impact on local observations during high ozone events) during the 2021 WiscoDISCO campaign.

**Specific comments:**

1/Page 6 lines 117-119: could authors show the forecast results and add a paragraph to explain how the ideal deployment period for the field campaign was chosen?

2/ Page 7 line 142: What's the saving frequency of the collected data?

3/ Page 10: Section 3 should be prior to the description of measurement location, deployment strategies and sampling.

4/ Page 27 lines 540-551: Authors should think to present these results in a separate section in the paper to demonstrate how the combination of measurements from UAS and Doppler lidar can be useful to characterize the lake breeze incursion. Conclusions should summarize the key supporting ideas you discussed throughout the work presented in the paper.

**Technical comments:**

1/ Page 2 line 38: please correct "on shoreline" by "on the shoreline".

2/ Page 2 line 63: please correct "night time"  by nighttime".

3/ Page 3 line 68: Please correct "Incorporation" by "The incorporation".

4/ Page 3 line 80: Please correct "up to planetary boundary layer" by " up to the planetary boundary layer".

5/ Page 4 line 104: Please correct "near to the WiDNR" by " near the WiDNR".

6/ Page 6 line 120: Please correct "cancelled" by " canceled".

---

## Referee Comment (RC2)

Cleary et al. reports the data set of lake breeze captured during the forecasted high ozone events at Chiwaukee Prairie State Natural Area in southeastern Wisconsin during the WiscoDISCO-21 campaign. The campaign used a fixed-wing (University of Colorado RAAVEN) and a multirotor (Purdue University DJI M210) UAS to measure temperature, humidity, 3-D wind and vertical profiles of temperature, humidity and ozone at two different altitude. The WiscoDISCO-21 was conducted in conjunction with Enhanced Ozone Monitoring plan from WI-DNR that included Doppler lidar wind profiler observations. This study demonstrates how UAS can be used to sample a complex circulation, and the obtained data are important for evaluating the characteristics of lake breeze incursion. Overall, the manuscript is well written, and I recommend publication of the current manuscript after the following revisions.

**General comment:**

Although a summary section has been included in the manuscript to summarize the 2021 WiscoDISCO field campaign, it is not very clear on the scientific importance of the data collected from the campaign. The authors should add a detail discussion here to show the scientific importance of the collected data. They should also highlight this in the abstract and introduction of the manuscript.

**Specific comments:**

Page 2, Line 46: List out the examples of precursors being emitted? General reader might not familiar with the precursors to ozone production.

Page 2, Line 50: Correct to Stanier et al. (2021).

Page 3, Line 89-90: Elaborate what are NO, NOy, NOx and VOC for non-expert reader.

Page 4, Line 101: Give the number for the highest ozone concentrations in the state.

Page 6, Line 117-119: Forecasts from both the WiDNR and Realtime Air Quality Modeling System (RAQMS) were used to select an ideal deployment period. I am curious to know how well the modeling forecast results agreed with the measurements.

Page 6, Table 1: Define what are flight Pattern A and B?

Page 8, Figure 2: How does the normalized probability being calculated in figure 2B (also for Fig.3)? The details should be stated in the caption.

Page 9, Figure 3: It is better to list the figure with alphabets rather than mentioning it from clockwise to top left.

Page 10, Line 161: Should be km hr$^{-1}$

Page 15, Figure 5: In this figure, it is clearer if the authors label the location of the bracket-mounted POM and iMET (similar to that in Fig.4).

Page 17, Figure 6: Please show the PANDORA and Doppler lidar inside the figure too.

Page 18, Line 333: The authors mentioned that 'A linear fit is then calculated to relate the VN-300 pressure and the difference between the VN-300 reported altitude and the autopilot reported

altitude. This pressure-dependent altitude correction is then applied to the VN-300-reported altitude to derive a final altitude'. Please show the figure or equation of linear fit that was used for correcting the altitude.

Page 20, Figure 7: The dash-line used to indicate one-to-one agreement is not clear. Suggest making it thicker or changing to another color.

Page 21, Line 383-384: What is the rationale for setting the limits to be 6% between RSS421 sensors, and 15% between the output RH value and the MHP-provided RH value?

Page 23, Line 454: Please specify how many flights.

---

## Author Response (AR1)

**Reviewer 1 Comments**

**Summary/recommendation**: This paper focuses on conducting a field campaign "The 2021 WiscoDISCO" that aims at capturing the elevated ozone concentration events resulting from the lake breeze circulation at Chiwaukee Prairie State Natural Area in Southeastern Wisconsin from May 21-26 2021. To do that, they deployed two Unmanned Aircraft Systems (UAS) platforms at two different altitudes (500 m above AGL for University of Colorado RAAVEN fixed wing UAS and 120 m above AGL for Purdue University DJI multirotor UAS) to measure temperature, humidity, 3D wind Sand and vertical profiles of temperature, humidity and ozone. Moreover, the 2021WiscoDISCO was conducted concurrently with the Enhanced Ozone Monitoring plan from WI-DNR that included a Doppler lidar wind profiler deployed on the roof of the Chiwaukee Prairie air mentoring station.

The scientific approach is sound and the work presented is substantial. However, the paper deserves more work before publication. I do recommend the publication of the current manuscript in ESSD journal if the authors consider these minor revisions in order to make a nice addition to the literature. I request that the authors consider the following points as they revise this manuscript:

**General comments:**

1/ The introduction focuses mainly on the goal of the 2021WiscoDISCO campaign and needs more details to explain the main reason for conducting this campaign and highlight the importance of capturing the breeze impact on local observations by adding more references. Also, authors need to highlight if models capture the lake breeze impact on shoreline ozone observations based on previous studies.

2/ Authors should add a section to explain how Scientists from other fields could benefit from the database created during this campaign. The risk is that the impact of the publication on a broader scientific community remains limited unless the authors put the paper into a wider perspective. Thus, the authors should add a section/paragraph dedicated to a scientific discussion including more references to previous studies to highlight the importance of the collected data (other than determining the breezes impact on local observations during high ozone events) during the 2021 WiscoDISCO campaign.

**Specific comments**:

1/Page 6 lines 117-119: could authors show the forecast results and add a paragraph to explain how the ideal deployment period for the field campaign was chosen?

2/ Page 7 line 142: What's the saving frequency of the collected data?

3/ Page 10: Section 3 should be prior to the description of measurement location, deployment strategies and sampling.

4/ Page 27 lines 540-551: Authors should think to present these results in a separate section in the paper to demonstrate how the combination of measurements from UAS and Doppler lidar can be useful to characterize the lake breeze incursion. Conclusions should summarize the key supporting ideas you discussed throughout the work presented in the paper.

**Authors' Response to Reviewer 1.**

Thank you for your thorough and helpful review of this manuscript. With respect to the comments in the review, all line numbers given for the revised manuscript refer to lines in the tracked-changes manuscript.

**General comments:**

1/The goals for the field campaign and background understanding of the lake breeze effect have been added to the revised manuscript. See P2 lines 39-51

2/Language to describe the utility of the dataset to the greater scientific community has been added to the revised manuscript. See P4 lines 111-124.

**Specific comments:**

1/ The selection of the time period for the campaign was dictated by capturing a combination of lake breeze and ozone events. We determined an acceptable window for operations from May 23-June 14 based on systems availability and the higher frequency of high ozone and lake breeze events occurring in this region during late spring/ early summer (see Cleary 2022 Atmospheric Environment, SI for a list of high ozone events for the years 2013-2019 at Chiwaukee Prairie). Once the window was approaching, the team used the RAQMS forecast model and consulted with the Wisconsin Department of Natural Resources Air Quality Division's meteorologist (who does internal ozone forecasting for the division) to decide on a "go time" to initiate deployment from all collaboration partners. The go time required evidence that synoptic flow would have a southerly component for a few days (normally brought about by a high-pressure system over the Ohio River Valley) with limited precipitation events. During the campaign, flight days were cancelled when operations would be in extreme weather or under conditions that were well outside of the goals (to capture lake breeze and to capture lake breeze with high ozone).

See P 18 lines 304-313.

2/ Page 7 line 142 (Now P21 lines 339-340): The request is to report the saving frequencies for instrumentation. The saving frequency for instrumentation for the RAAVEN is given in lines 357-365 and the saving frequency for instrumentation on the M210 is given in lines 492-493 and units were changed from intervals in s to Hz.

3/ Section 3 has been moved. Please see revised manuscript.

4/ Thank you for this suggestion – we now have a separate section for interpreted results starting on Line 584 on P32.

P 32 lines 584-608 now include interpreted results.

**Technical comments**

1/ Page 2 line 38: please correct "on shoreline" by "on the shoreline". Done

2/ Page 2 line 63: (now line 78) please correct "night time" by nighttime".  Done

3/ Page 3 line 68: (now line 83) Please correct "Incorporation" by "The incorporation". Done

4/ Page 3 line 80: (now P4 line 95-96) Please correct "up to planetary boundary layer" by " up to the planetary boundary layer". Done

5/ Page 4 line 104: (now P16 line 288-289) Please correct "near to the WiDNR" by " near the WiDNR". Done

6/ Page 6 line 120: (now P 18 line 313) Please correct "cancelled" by " canceled". Done
* * *
**Reviewer 2 Comments**

Cleary et al. reports the data set of lake breeze captured during the forecasted high ozone events at Chiwaukee Prairie State Natural Area in southeastern Wisconsin during the WiscoDISCO-21 campaign. The campaign used a fixed-wing (University of Colorado RAAVEN) and a multirotor (Purdue University DJI M210) UAS to measure temperature, humidity, 3-D wind and vertical profiles of temperature, humidity and ozone at two different altitude. The WiscoDISCO-21 was conducted in conjunction with Enhanced Ozone Monitoring plan from WI-DNR that included Doppler lidar wind profiler observations. This study demonstrates how UAS can be used to sample a complex circulation, and the obtained data are important for evaluating the characteristics of lake breeze incursion. Overall, the manuscript is well written, and I recommend publication of the current manuscript after the following revisions.

**General comment:** Although a summary section has been included in the manuscript to summarize the 2021 WiscoDISCO field campaign, it is not very clear on the scientific importance of the data collected from the campaign. The authors should add a detail discussion here to show the scientific importance of the collected data. They should also highlight this in the abstract and introduction of the manuscript.

**Specific comments**:

Page 2, Line 46: List out the examples of precursors being emitted? General reader might not familiar with the precursors to ozone production.

Page 2, Line 50: Correct to Stanier et al. (2021). Page 3, Line 89-90: Elaborate what are NO, NOy, NOx and VOC for non-expert reader.

Page 4, Line 101: Give the number for the highest ozone concentrations in the state.

Page 6, Line 117-119: Forecasts from both the WiDNR and Realtime Air Quality Modeling System (RAQMS) were used to select an ideal deployment period. I am curious to know how well the modeling forecast results agreed with the measurements.

Page 6, Table 1: Define what are flight Pattern A and B?

Page 8, Figure 2: How does the normalized probability being calculated in figure 2B (also for Fig.3)? The details should be stated in the caption.

Page 9, Figure 3: It is better to list the figure with alphabets rather than mentioning it from clockwise to top left.

Page 10, Line 161: Should be km hr-1 Page 15, Figure 5: In this figure, it is clearer if the authors label the location of the bracket-mounted POM and iMET (similar to that in Fig.4).

Page 17, Figure 6: Please show the PANDORA and Doppler lidar inside the figure too.

Page 18, Line 333: The authors mentioned that 'A linear fit is then calculated to relate the VN-300 pressure and the difference between the VN-300 reported altitude and the autopilot reported altitude. This pressure-dependent altitude correction is then applied to the VN-300-reported altitude to derive a final altitude'. Please show the figure or equation of linear fit that was used for correcting the altitude.

Page 20, Figure 7: The dash-line used to indicate one-to-one agreement is not clear. Suggest making it thicker or changing to another color.

Page 21, Line 383-384: What is the rationale for setting the limits to be 6% between RSS421 sensors, and 15% between the output RH value and the MHP-provided RH value?

Page 23, Line 454: Please specify how many flights.
* * *
**Authors' Response to Reviewer 2**

We thank the reviewer for their thorough examination of this manuscript.

In response to the general comment: the data from this field campaign is targeted in better understanding the role of lake breeze on shoreline ozone concentrations. The vertical structure obtained from the measurements will allow investigations into the dimensions of the stable boundary layer, the height of which can be modeled using a Richardson Number. Such investigations can be used in comparison to current weather forecasting models to test the validity in a shoreline environment. The vertical profile of ozone and meteorological variables allows for a nuanced understanding of the changes to stratification and vertical mixing occur as an air mass moves from over-water to over-land and how the coastal interface plays a role in pollutant mixing within the marine layer and lake breeze movement. Understanding the vertical and horizontal dimensions of the lake breeze phenomena can also inform model resolution improvements in this area impacted by high ozone, where 1.3 km grid scale models are only now starting to capture the lake breeze adequately. See revised manuscript P4-5, lines 111-124.

All **specific comments** have been addressed in the revised manuscript.

P 2, Line 46, (now lines 58-59): "volatile organic compounds (VOCs) and nitrogen oxides (NOx)" is added

P 2, line 50, (now P2 line 63): Deleted comma.

P 3, line 89-90, (now P 4-5 lines 111-125): Additional definitions given along with how the measurements can be used.

P 4, line 101, (now P16 lines 283-285): The 8-hour design value for the site is given along with the 8-hour federal ozone standard.

P 6: line 117-119: (Figure 5 has been added, now on P19): The shoreline monitor ozone was not modeled well, although the model predicted high ozone over the lake and the meteorological forecast

suggested the possibility of a shallow lake breeze on May 22$^{nd}$ and 24$^{th}$. Lines 304-313 on P18 accompany this new figure.

P 6, Table 1 (now P 19): language in the caption indicates patterns A and B are shown in Figure 2. Figure 2 caption also gives the color schemes for Pattern A and B.

Page 8, Figure 2 (Now Figure 6, P21): Figures 6 and 7 have "normalized probabilities" that are constructed so that the integral curve is equal to 1. Here, the value provided for a given bin is the number of elements in a given bin divided by the total number of elements in the input data. See figure caption.

Page 9 Figure 3 (Now Figure 7, P23): The figure has been updated to include the letters, as recommended.

Page 10, line 161 (now P5 line 134): units are corrected.

Page 15, Figure 5 (Now Figure 2, P12): Changed.

Page 17, Figure 6 (Now Figure 3, P15): Changed.

Page 18, Line 333 (Now P 24, lines 384-387): This text describes the procedure applied, not a single equation or relationship for the entire dataset. This procedure is applied independently to each flight to account for differences resulting from weather conditions and location-specific GPS offsets for a given flight that may impact the accuracy of the altitude estimate. Therefore, it is not practical to show a single figure or single equation, as there will be slight differences from one flight to the next.

Page 20, Figure 7 (Now Figure 8, P 26): We have updated the figure and associated caption to make the one-to-one line red.

Page 21, Line 383-384 (Now P 27 lines 439-445): The relative humidity values from the multihole pressure probe (MHP) are significantly impacted by the exposure of that sensor to sunlight and the associated impact on sensor temperature. This is not corrected for, resulting in large fluctuations in the RH values at times. As a result, this measurement (from the MHP) only provides a reality check to ensure that the RSS-421 is reporting something that is in line with the general reality, and therefore such a large offset (15%) is allowed. The more important comparison is between the two RSS-421 sensors, which should agree much more closely, as they are the same sensor type, and are mounted within close proximity of one another. Historical comparisons between the sensors across a wide variety of environmental regimes have led us to understand that 6% is an acceptable threshold for this comparison.

Page 23, Line 454 (now P 29 line 514): 2 flights did not recover iMET data because of a loss in battery power.